neuroscience/behaviour

domestication, brain, cerebellum, stereology, allometry

**Author for correspondence:**
Andrew N. Iwaniuk
e-mail: andrew.iwaniuk@uleth.ca

# The cerebellar anatomy of red junglefowl and white leghorn chickens: insights into the effects of domestication on the cerebellum

Kelsey J. Racicot[1], Christina Popic[1], Felipe Cunha[1], Dominic Wright[2], Rie Henriksen[2] and Andrew N. Iwaniuk[1]

[1]Department of Neuroscience, University of Lethbridge, Lethbridge, Alberta, Canada T1K3M4
[2]AVIAN Behavioural Genomics and Physiology Group, IFM Biology, Linköping University, Linköping 58183, Sweden

ANI, 0000-0001-9273-3655

Domestication is the process by which wild organisms become adapted for human use. Many phenotypic changes are associated with animal domestication, including decreases in brain and brain region sizes. In contrast with this general pattern, the chicken has a larger cerebellum compared with the wild red junglefowl, but what neuroanatomical changes are responsible for this difference have yet to be investigated. Here, we quantified cell layer volumes, neuron numbers and neuron sizes in the cerebella of chickens and junglefowl. Chickens have larger, more folded cerebella with more and larger granule cells than junglefowl, but neuron numbers and cerebellar folding were proportional to cerebellum size. However, chickens do have relatively larger granule cell layer volumes and relatively larger granule cells than junglefowl. Thus, the chicken cerebellum can be considered a scaled-up version of the junglefowl cerebellum, but with enlarged granule cells. The combination of scaling neuron number and disproportionate enlargement of cell bodies partially supports a recent theory that domestication does not affect neuronal density within brain regions. Whether the neuroanatomical changes we observed are typical of domestication or not requires similar quantitative analyses in other domesticated species and across multiple brain regions.

# 1. Introduction

Domestication is an evolutionary process in which wild animals are captive-bred and selected to possess traits beneficial for humans [1,2]. Domesticated animals are bred for a myriad of reasons, some of which include food (e.g. milk, meat), companionship, work (e.g. pack horses, guide dogs), research (e.g. laboratory rats and mice) and other resources (e.g. leather, fur). Despite being bred for a range of different purposes, domesticates possess a host of behavioural, physiological and morphological characters that differ from their wild counterparts, which are collectively referred to as 'domestication syndrome' [3]. Domestication syndrome, especially in mammals, encompasses changes in many traits, such as a decrease in body and brain size, alterations in coloration, reduced organ size, docility towards humans, progenesis and neoteny [3–5]. With respect to the brain, domesticated strains typically have relatively smaller brains than their wild counterparts [6]. Volumetric decreases are widespread across olfactory, visual, auditory, limbic and motor regions in mammals [6–8] and birds [9–13], with all of them contributing to differences in relative brain size between wild and domesticated strains of the same species.

In some bird species, however, domesticated strains contradict this general pattern by having larger, rather than smaller, brain regions compared with wild strains. For example, homing pigeons (*Columba livia*) have relatively larger hippocampal formations and olfactory bulbs compared with wild rock doves [9,13,14], presumably the result of centuries of selective breeding for homing behaviour [13]. More recently, Henriksen *et al*. [15] also found that white leghorn chickens (*Gallus gallus domesticus*) have relatively larger cerebella than their wild progenitor, the red junglefowl. The cerebellar enlargement in chickens was associated with decreased time spent brooding and variation among several candidate genes, at least one of which regulates cerebellar development (*FGF9*, [15]). However, quantitative neuroanatomical data are lacking. That is, whether chickens have more and/or larger cerebellar neurons than junglefowl is unknown. Quantifying cerebellar anatomy of chickens and junglefowl would provide insights into what mechanisms are responsible for cerebellar enlargement in chickens. For example, if the enlargement is primarily due to extending granule cell neurogenesis, then chickens would have disproportionately more granule cells than junglefowl. Further, a comparison of neuron numbers and sizes between chickens and junglefowl would be the first detailed quantitative analysis of the brains of domesticated versus wild strains in any species.

Here, we provide a detailed quantitative analysis of the cerebellum in both white leghorn chickens and junglefowl to determine what is responsible for cerebellar enlargement in chickens. More specifically, we quantified volumes of cerebellar cortex layers and the number and sizes of the two major neuronal populations in the cerebellum: Purkinje cells and granule cells. Based on comparative studies of brain anatomy, we hypothesized that cerebellar enlargement in chickens is primarily due to the addition of neurons, and, therefore, chickens would have more granule and Purkinje cells than junglefowl. Cerebellar foliation, the degree of folding of the cerebellar cortex, varies with brain and cerebellum size in birds [16,17] so we also predicted that the larger chicken cerebellum would be more folded than that of the junglefowl. Overall, this represents the most comprehensive quantitative comparison of brain region anatomy between a wild-type and domesticated strain that can shed some light on how domestication shapes brain anatomy.

# 2. Material and methods

## 2.1. Animals

Six male white leghorn chicken (*G. gallus domesticus*) and six male red junglefowl (*G. gallus*) specimens (hereafter 'chickens' and 'junglefowl') were obtained from breeding colonies maintained at Linköping University (Linköping, Sweden). The junglefowl were derived from a Swedish zoo population and had been kept at the research facility since 1998. The domesticated chickens originated from a selection line, SLU13, bred at the Swedish University of Agricultural Sciences. The background of these lines has been described in more detail by Schütz & Jensen [18]. Both lines had been continuously bred based on pedigree in order to sustain genetic diversity. At the research station, the birds had access to commercial chicken feed and freshwater ad libitum and were exposed to a 12 : 12 h light/dark cycle. The birds were kept in large groups in pens, separated by sex and strain. Housing pens measured 2.5 × 3 m and comprised three separate levels equipped with perches. All birds had access to outdoor pens of the same size. The study was approved by the local Ethical Committee of the Swedish National Board

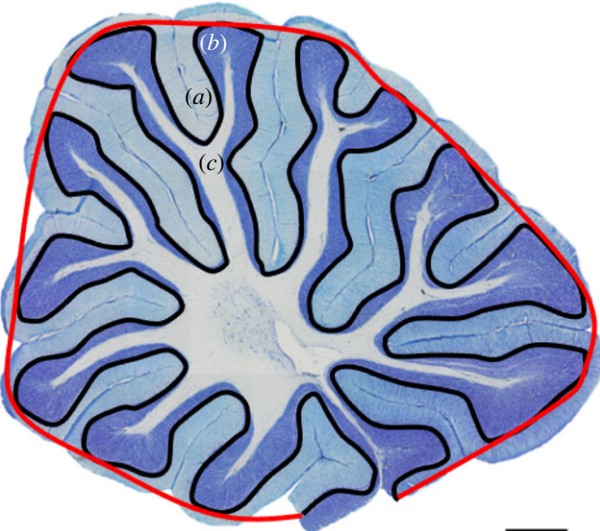

**Figure 1.** Midsagittal Nissl-stained section of a red junglefowl (*Gallus gallus*) cerebellum. The red line indicates the Purkinje cell layer envelope and the black line indicates the total Purkinje cell layer length. The cerebellar foliation index (CFI) was calculated by dividing the length of the black line by the length of the red line (*sensu* [16]). (*a*) The molecular layer, (*b*) the granule cell layer and (*c*) the white matter. Scale bar, 1 mm.

for Laboratory Animals (ethical permit Dnr 50-13). All individuals were culled by cervical neck dislocation followed by decapitation (as per the ethical permit), and their brains were extracted from the skull immediately after and immersed in 4% buffered paraformaldehyde (PFA).

## 2.2. Histology

After fixation in formaldehyde for several weeks, the cerebella were dissected from the rest of the brain by carefully cutting the cerebellar peduncles until fully detached. The cerebella were placed in 30% sucrose + 0.1 M phosphate-buffered saline (PBS) until sunken to ensure cryoprotection. Once sunken, the cerebella were embedded in a gelatin/sucrose medium, post-fixed in 20% sucrose + 4% PFA (pH = 7.4) for 2–3 h, and then placed in 20% sucrose + 0.1 M PBS overnight. The embedded cerebella were then sectioned on a freezing stage microtome into 40 µm thick sections in the sagittal plane. Every other section (1 : 2 series) was mounted onto 1% gelatin and 0.2% chromium potassium phosphate subbed slides. They were then stained with thionin acetate (AcrosOrganic, 78338-22-4) for Nissl substance, dehydrated with a graded ethanol series, de-fatted with Hemo-De (Fisher Scientific, HD-150), then coverslipped with Permount (Fisher Scientific, SP15-500) and left to dry for at least one week before stereological analysis.

## 2.3. Neuroanatomical measurements

We measured the volume of each of the cell layers (molecular, granular, white matter and total cerebellum (CB) volume) using the Cavalieri estimator in StereoInvestigator (MBF Bioscience, Williston, VT, USA). We measured between 15 and 20 sections per cerebellum and a grid size of 400 × 400 µm for all three layers measured: molecular, granule cell and white matter (figure 1). The Purkinje cell layer volume was not measured because it is only a single cell layer thick and there can be large gaps between Purkinje cells in sagittal sections, especially in the sulci of the cerebellar folia. Also, due to difficulties in accurately defining the cerebellar nuclei in sagittal sections, the cerebellar nuclei were included in the white matter volume. Coefficients of error (CEs, Gunderson, $m = 1$) were all less than or equal to 0.005 for total CB volume, less than or equal to 0.006 for molecular layer volume, less than or equal to 0.008 for granular layer volume and less than or equal to 0.012 for white matter volume.

The number of Purkinje cells was estimated using a 20× lens on a Zeiss Axiocam Imager MRm microscope (Carl Zeiss, MicroImaging GmBH, Germany) and the optical fractionator tool in StereoInvestigator (MBF Bioscience, Williston, VT, USA). We sampled 10–11 sections per animal with a counting frame of 80 × 80 µm, a sampling grid of 400 × 400 µm and a dissector height of 15 µm. Purkinje cells were identified by their teardrop shape, location within the Purkinje cell layer and

visible axons and dendrites extending into the adjacent molecular and granule cell layers. Only cells with an intact cell membrane and visible nucleus were counted. The number of cells counted ranged from 375 to 621 and the CEs were all less than or equal to 0.05.

Purkinje cell soma size was determined with the nucleator tool in conjunction with the optical fractionator to ensure that cells were randomly selected within and across sections [19]. We used the nucleator with four rays to estimate the cross-sectional area ($\mu m^2$) of Purkinje cells, with the nucleus as the centre. The same frame size as cell counting was used ($80 \times 80 \, \mu m$), and the sampling grid size used was $800 \times 800 \, \mu m$. This method was used to randomly select Purkinje cells throughout the sections and the number of cells measured per specimen ranged from 58 to 84. Purkinje cells were identified in the same way as they were for counting, and only cells with a clearly defined cell membrane and nucleus were measured. The CEs for Purkinje cell soma areas were all less than or equal to 0.008.

The number of granule cells in each cerebellum was also estimated using the optical fractionator. Using a 100× immersion oil lens, we counted granule cells in 9–10 sections per cerebellum with a counting frame of $10 \times 10 \, \mu m$ and a sampling grid of $900 \times 900 \, \mu m$. We used a dissector height of $4 \, \mu m$, and a top and bottom guard zone of $4 \, \mu m$, which resulted in an equal distribution of cells throughout the counting volume [20]. Granule cells were identified by their round shape and location in the granule cell layer. Only cells with an intact cell membrane and nucleus were counted. Because granule cells are densely packed, we made sure to only count one cell at a time by identifying the nucleus in each granule cell to avoid counting overlapping cells as one cell. Granule cells are not always distinguishable from non-neuronal cells (e.g. glia) in Nissl-stained tissue. Thus, our granule cell counts probably represent an over-estimation of granule cell numbers compared with that obtained from isotropic fractionation studies that use NeuN immunohistochemistry [21]. CEs for the granule cell counts were all less than or equal to 0.06.

Again, soma size was determined with the nucleator tool in conjunction with the optical fractionator [19]. The area of each granule cell soma was estimated from a random selection of 100–150 morphologically intact cells to produce an average estimate cell soma area. The same frame size ($10 \times 10 \, \mu m$) and grid size ($900 \times 900 \, \mu m$) were used as for counting. Granule cells were identified in the same manner as they were for counting, with an intact cell membrane and visible nucleus and we used a nucleator with four rays to estimate the area ($\mu m^2$) of each cell soma. All CEs for area were less than or equal to 0.002.

Last, to quantify the degree of folding, we used the contour tool in StereoInvestigator to measure the total length of the Purkinje cell layer and the total length of the cerebellum envelope in each midsagittal section (figure 1). By dividing the total length of the Purkinje layer by the length of the envelope, we get a value we call the cerebellar foliation index (CFI) [16]. The CFI gives us a ratio to estimate the degree of folding, a method similar to measuring the cortical gyrification in mammals [22].

## 2.4. Statistical analysis

We used one-way analyses of variance (ANOVAs) to test for absolute differences between the chickens and junglefowl for the following variables: body mass, brain mass, cerebellar mass, cerebellar volume, CFI, Purkinje cell number, Purkinje soma size, granule cell number and granule soma size. We also used one-way analyses of covariance (ANCOVAs) to test for significant differences in allometric relationships among the following comparisons: CB mass against the rest of the brain mass; each layer against total CB volume minus that layer; Purkinje cell size and number against total CB volume; granule cell size and number against total CB volume; and CFI against total CB volume. All data used in the ANCOVAs were log-transformed before statistical analyses to ensure normalization.

# 3. Results

Chickens have significantly larger body masses, brain masses, cerebellar masses and cerebellar volumes than junglefowl (figure 2 and table 1). There was not, however, a clear difference in relative brain mass between chickens and junglefowl. Our statistical analysis revealed a significant difference in slopes between the two strains ($F$(slope) = 7.54 d.f. = 1, 8, $p = 0.03$; figure 3a), but this largely arises from a lack of a significant allometric relationship between body and brain size in the junglefowl ($F = 0.69$, d.f. = 1, 4, $p = 0.45$).

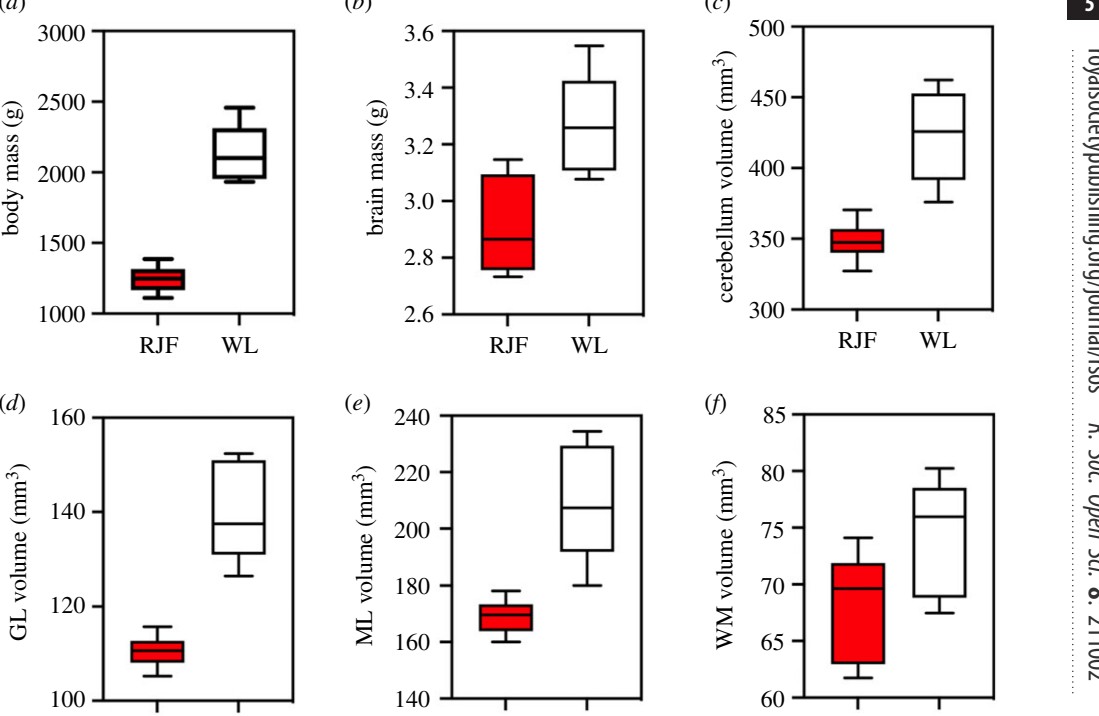

**Figure 2.** Boxplots (min-max) of absolute values of the following: (*a*) body mass (g); (*b*) brain mass (g); (*c*) cerebellum volume (mm³); (*d*) granule cell layer (GL) volume (mm³); (*e*) molecular layer (ML) volume (mm³) and (*f*) white matter (WM) volume (mm³). In all graphs, red junglefowl (RJF) are shown in red and white leghorn chickens (WL) are shown in white.

Relative to the rest of the brain, the cerebellum is also significantly larger in chickens (+16.6%) than in junglefowl (ANCOVA: $F$(strain) = 8.53, d.f. = 1, 8, $p = 0.02$, other effects were not significant; figure 3*b*). Chickens also had larger granule cell (figure 2*d*) and molecular layer volumes (figure 2*e*) than junglefowl, but the two strains did not differ in white matter volumes (figure 2*f* and table 1). Relative to the rest of the cerebellum (figure 3*c–e*), only the granule cell layer was disproportionately larger (+11.6%) in chickens (table 2).

As shown in figure 4, the chicken cerebellum appears to be not only larger, but also more folded than that of junglefowl. This observation was supported by our statistical analysis; the chicken cerebellum is significantly more folded than that of junglefowl (figure 4*c* and table 1). The chicken cerebellum is not more folded than what is expected based on the cerebellum size (figure 4*d*), as indicated by the lack of a significant difference in CFI relative to cerebellar volume (table 3). It should, however, be noted that chickens had more variable CFI measurements than junglefowl (figure 4*c,d*).

Chickens had significantly more Purkinje cells than junglefowl (figure 5*a* and table 1), but not relative to cerebellar volume (table 3), indicating that the number of Purkinje cells is proportional to cerebellum size (figure 5*b*). Purkinje cell soma size differed significantly between chickens and junglefowl in absolute terms, but the effect size was marginal (figure 5*c* and table 1). Relative to the cerebellum volume, no significant difference in Purkinje cell sizes was detected between the two strains (figure 5*d* and table 3).

Last, chickens have significantly more (figure 5*e*) and larger (5*g*) granule cells than junglefowl (table 1). The relative number of granule cells did not differ significantly between the two strains (figure 5*f* and table 3), but the size of the granule cells did (5*h* and table 3). Thus, chickens have disproportionately larger granule cells (+16%) than junglefowl.

## 4. Discussion

Our results corroborate previous findings that chickens have larger cerebella, in both relative and absolute terms, than junglefowl [15]. In addition, chickens have more folded cerebella with larger granule cells and more Purkinje cells than junglefowl. Relative to the total cerebellum size, chickens have disproportionately larger granule cell layer volumes and larger granule cells than junglefowl, but

**Table 1.** Means and standard deviations (s.d.) for 12 neuroanatomical measurements taken of junglefowl and chicken brains and the results of analyses of variance tests (all d.f. = 1, 11). Significant p-values are shown in italics. The strains are abbreviated as WL, white leghorn chicken; RJF, red junglefowl. '%diff' indicates the percentage difference of the chickens relative to the junglefowl.

| measurements | red junglefowl | | | white leghorn chicken | | | F | p-value | %diff |
|---|---|---|---|---|---|---|---|---|---|
| | n | mean | s.d. | n | mean | s.d. | | | |
| body mass (kg) | 6 | 1.24 | 0.09 | 6 | 2.14 | 0.20 | 119.53 | *<0.01* | +72.6 |
| brain mass (mg) | 6 | 2908.65 | 168.69 | 6 | 3274.17 | 182.25 | 13.26 | *<0.01* | +12.6 |
| cerebellar mass (mg) | 6 | 385.87 | 19.05 | 6 | 471.42 | 41.22 | 24.26 | *<0.01* | +22.2 |
| cerebellar volume (mm$^3$) | 6 | 348.23 | 13.98 | 6 | 422.67 | 33.00 | 28.15 | *<0.01* | +21.4 |
| cerebellar foliation index | 6 | 3.42 | 0.12 | 6 | 3.83 | 0.29 | 10.84 | *<0.01* | +12.0 |
| granule cell number | 6 | $1.86 \times 10^8$ | $2.59 \times 10^7$ | 6 | $2.28 \times 10^8$ | $4.69 \times 10^7$ | 4.37 | 0.06 | +22.6 |
| Granule soma size (μm$^2$) | 6 | 16.82 | 0.72 | 6 | 18.96 | 0.68 | 27.73 | *<0.01* | +12.7 |
| Purkinje cell number | 6 | $2.89 \times 10^5$ | $2.34 \times 10^4$ | 6 | $3.63 \times 10^5$ | $2.43 \times 10^4$ | 28.58 | *<0.01* | +25.6 |
| Purkinje soma size (μm$^2$) | 6 | 378.57 | 23.44 | 6 | 352.58 | 16.05 | 5.02 | *0.049* | −6.9 |
| molecular layer volume (mm$^3$) | 6 | 169.07 | 6.20 | 6 | 208.77 | 20.13 | 23.92 | *<0.01* | +23.5 |
| granule cell layer volume (mm$^3$) | 6 | 110.50 | 3.47 | 6 | 139.45 | 10.48 | 48.34 | *<0.01* | +26.2 |
| white matter volume (mm$^3$) | 6 | 68.27 | 4.77 | 6 | 74.48 | 5.02 | 4.71 | 0.06 | +9.1 |

not more numerous granule cells. Thus, the enlargement of the chicken cerebellum arises from both coordinated increases in neuron numbers and a disproportional enlargement of cells within the granule cell layer.

One limitation of the present study is that the junglefowl were captive-bred, so although they resemble the wild-type, captive breeding may have affected their neuroanatomy. Captive-bred ducks have relatively smaller brains than their wild counterparts [23] and similar effects have been reported in a range of other species [24–26]. Captivity can alter the size of the hippocampal formation in birds [27,28], but whether captive breeding, in the absence of intentional selection for phenotypic traits, is sufficient to drive a change in the cerebellum is entirely unknown. That said, both junglefowl and chickens are captive-bred, so if junglefowl had undergone a reduction in the cerebellum size, this would indicate that the chickens had still enlarged the cerebellum. The question is whether the chicken is more representative of the ancestral state or not. Sourcing truly wild junglefowl that have not interbred with chickens is not a trivial task, but some insights could be gleaned by examining the neuroanatomy of feral chickens [29,30].

It is also worth noting that the current analysis focused only on males. There is the potential for breed-specific differences in cerebellar anatomy to vary between the sexes, as recently demonstrated in artificially selected lines of Japanese quail (*Coturnix japonica* [31]). Further, there is also the potential for varying degrees of differences in neuron sizes and numbers between junglefowl and chickens across other brain regions. A broader range of quantitative measurements of both sexes and strains is needed to gain insights into what makes a chicken brain different from that of a junglefowl brain.

In contrast with many domesticate–wild pair comparisons [6,9–12], we did not find a significant difference in the relative brain size between chickens and junglefowl. This is unexpected, given that the difference in absolute body mass is much larger than that of brain mass (table 1). Chickens also

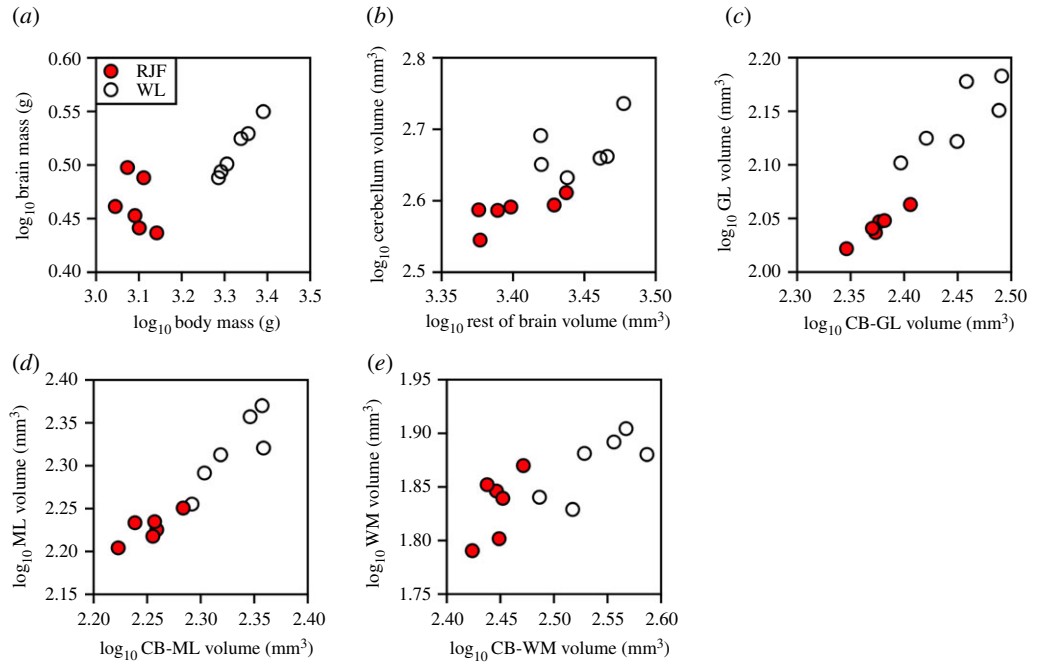

**Figure 3.** Scatterplots of brain mass plotted against body mass and the volumetric measurements of the specimens examined, with all variables $log_{10}$-transformed. The individual scatterplots are as follows: (*a*) brain mass (g) against body mass (g); (*b*) cerebellar volume (mm$^3$) against the rest of the brain volume (mm$^3$); (*c*) granule cell layer (GL) volume (mm$^3$) against cerebellum (CB) minus granule cell layer volume (mm$^3$); (*d*) molecular layer (ML) volume (mm$^3$) against cerebellum minus molecular layer volume (mm$^3$) and (*e*) white matter (WM) volume (mm$^3$) against cerebellum minus white matter volume (mm$^3$). In all graphs, red junglefowl (RJF) are shown in red circles and white leghorn chickens (WL) are shown in white circles.

**Table 2.** Results from one-way analyses of covariance (ANCOVAs) of strain, total cerebellar volume minus layer volume and their interaction on three volumetric measurements: granule cell layer, molecular layer and white matter. Significant effects are shown in italics and a Tukey HSD *post hoc* test provided where strain is significant (WL, white leghorn chicken; RJF, red junglefowl).

| measurement | strain | | | | total volume – layer volume | | | interaction | | |
|---|---|---|---|---|---|---|---|---|---|---|
| | *F* | d.f. | *p*-value | Tukey HSD | *F* | d.f. | *p*-value | *F* | d.f. | *p*-value |
| granule cell layer | 7.64 | 1, 11 | *0.03* | WL > RJF | 13.00 | 1, 11 | *<0.01* | 0.002 | 1, 11 | 0.97 |
| molecular layer | 0.64 | 1, 11 | 0.45 | | 15.91 | 1, 11 | *<0.01* | 1.89 | 1, 11 | 0.21 |
| white matter | 2.19 | 1, 11 | 0.18 | | 7.41 | 1, 11 | *0.03* | 1.00 | 1, 11 | 0.35 |

experience far greater somatic growth than neural growth throughout post-hatching development [15]. Although domesticated turkeys have much smaller brains, relative to body size, than wild turkeys (*Melagris gallopavo*, [11]), the difference in mallard ducks (*Anas platyrhynchos*) is much smaller [12] and there is no clear difference in the relative brain size between rock doves and domesticated pigeons [13]. Similar variation across species in the amount that relative brain size decreases with domestication also occurs in mammals [6]. The lack of a difference between junglefowl and chickens reported here might reflect a smaller sample size than previous studies [9–12] or a smaller difference in the relative brain size between chickens and junglefowl than expected. Ideally, this should be probed further with a much larger sample size and the inclusion of a variety of chicken breeds.

Previous work on allometric scaling relationships of neuron numbers in carnivoran mammals suggested that domesticates do not differ in their scaling relationships compared with their wild counterparts [32]. Despite the lack of an explicit comparison of domesticated versus wild strains within a species in that study, our results generally support this claim; the chicken cerebellum is primarily a scaled-up version of the junglefowl cerebellum. The chicken has larger volumes of the

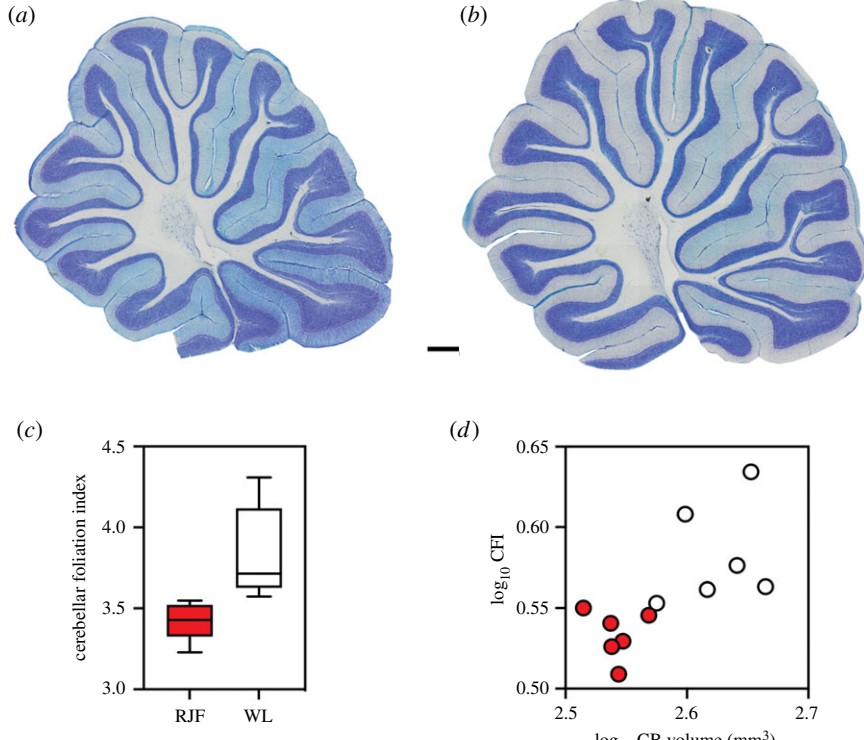

**Figure 4.** Midsagittal Nissl-stained sections of red junglefowl (*G. gallus*) cerebellum (*a*) and white leghorn chicken (*G. gallus domesticus*) cerebellum (*b*). Scale bar, 1 mm. (*c*) A boxplot (min-max) of the cerebellar foliation index (CFI) illustrating the difference between junglefowl (RJF) and chickens (WL). (*d*) A log–log scatterplot of CFI plotted against cerebellum (CB) volume with red junglefowl shown in red circles and white leghorn chickens shown in white circles.

**Table 3.** Results from one-way analysis of covariance (ANCOVAs) of strain, cerebellar volume and their interaction on five measurements: number of granule cells, granule cell size (μm$^2$), number of Purkinje cells, Purkinje cell size (μm$^2$) and CFI. Significant effects are shown in italics and *post hoc* tests provided where strain is significant (WL, white leghorn chicken; RJF, red junglefowl).

| measurement | strain | | | | CB volume | | | Interaction | | |
|---|---|---|---|---|---|---|---|---|---|---|
| | *F* | d.f. | *p*-value | Tukey HSD | *F* | d.f. | *p*-value | *F* | d.f. | *p*-value |
| number of granule cells | 0.31 | 1, 8 | 0.59 | | 2.73 | 1, 8 | 0.14 | 0.63 | 1, 8 | 0.45 |
| granule cell size | 6.91 | 1, 8 | *0.03* | WL > RJF | 0.37 | 1, 8 | 0.56 | 0.11 | 1, 8 | 0.75 |
| number of Purkinje cells | 1.91 | 1, 8 | 0.20 | | 0.93 | 1, 8 | 0.36 | 1.36 | 1, 8 | 0.28 |
| Purkinje cell size | 0.35 | 1, 8 | 0.57 | | 0.00 | 1, 8 | 0.99 | 0.50 | 1, 8 | 0.50 |
| cerebellar foliation index | 1.50 | 1, 8 | 0.26 | | 0.03 | 1, 8 | 0.87 | 0.26 | 1, 8 | 0.63 |

three layers and more Purkinje and granule cells than the junglefowl, and many of these differences are of similar magnitude (table 1). By contrast, there were few differences in relative volumes and none in relative neuron numbers, indicating that chickens do not have proportionally more neurons than junglefowl and that the changes in the absolute size follow similar scaling 'rules' in both strains. Thus, the chicken cerebellum is, in many respects, a scaled-up version of a junglefowl cerebellum. This parallels recent findings across galliform species [17] in which there are strong allometric relationships for all volumes and cell counts with overall cerebellum size (all $r_s^2 > 0.9$). Volumes of cerebellar cortex layers are also strongly correlated with total cerebellum volume in quail strains differentially selected for divergent reproductive investment [31]. Together, these results indicate that enlarging the cerebellum is achieved through increases across layers of cerebellar cortex and neuronal populations that adhere to intraspecific allometry.

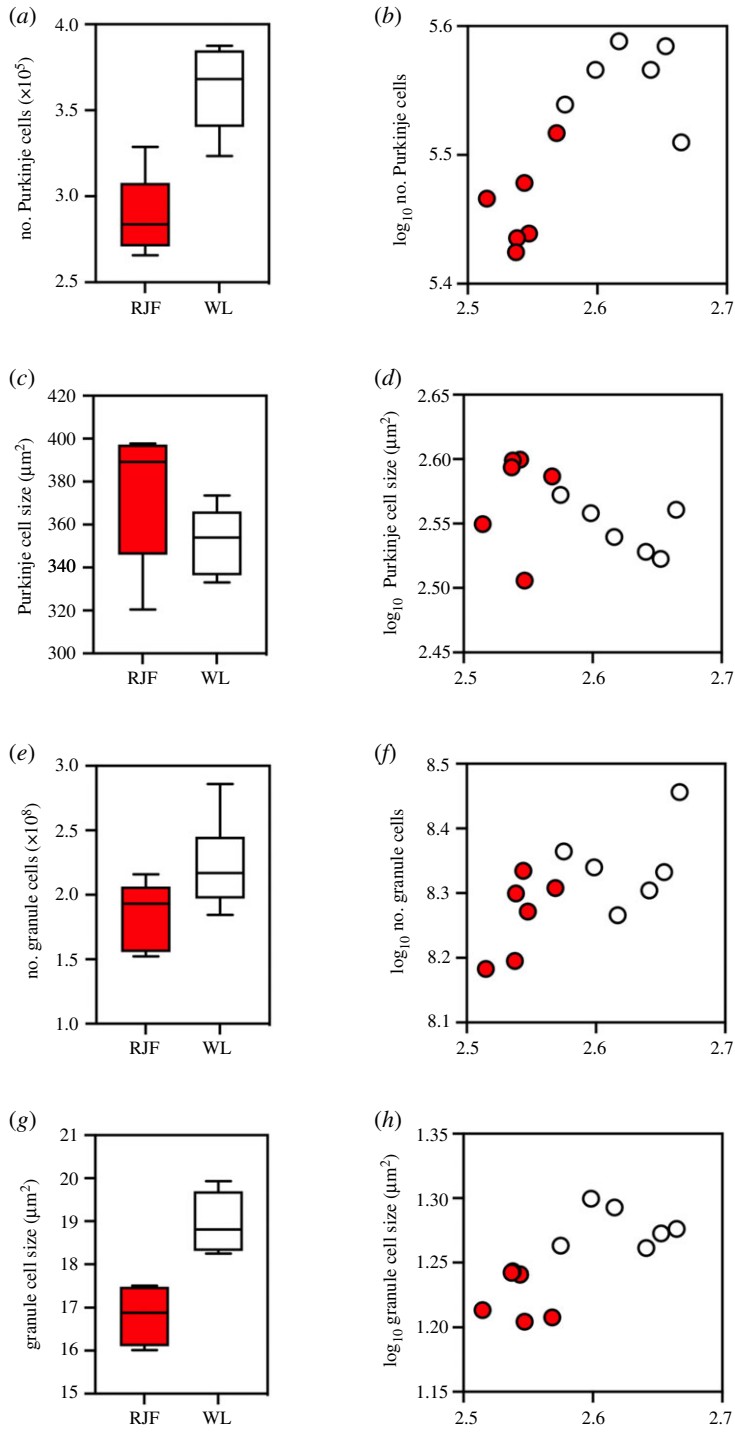

**Figure 5.** Boxplots (min–max) and scatterplots of numbers and sizes of Purkinje and granule cells. In all graphs, red junglefowl (RJF) are shown in red and white leghorn chickens (WL) are shown in white. The graphs are as follows: (*a*) boxplots of Purkinje cell numbers; (*b*) log–log scatterplot of Purkinje cell numbers against cerebellum (CB) volume; (*c*) boxplots of Purkinje cell soma sizes; (*d*) log–log scatterplot of Purkinje cell soma size against CB volume; (*e*) boxplots of the number of granule cells; (*f*) log–log scatterplot of number of granule cells against CB volume; (*g*) boxplots of granule cell soma sizes and (*h*) log–log scatterplot of granule cell soma size against CB volume.

Despite this overall pattern of conserved scaling, chickens have disproportionately larger granule cell layer volumes than junglefowl. As shown in figure 3*c*, the scaling relationship between granule cell layer volume and the rest of the cerebellum is similar between chickens and junglefowl, but they differ in intercept (table 2). Rather than this difference in granule cell layer volume being driven by

neuron numbers, it could be driven by either increases in neuropil or cell sizes. Due to the density of granule cells, we were unable to measure neuropil effectively, but the average granule cell size is larger in chickens than in junglefowl in both absolute and relative terms (figure 5g,h). Thus, chickens have proportionately larger granule cell layers with larger cells than junglefowl, in contrast with the highly conserved model proposed by Jardim-Messeder *et al.* [32]. Soma size is positively correlated with the number and size of organelles, with larger cells supporting increased protein synthesis [33]. In neurons, soma size also reflects physiological properties. Indeed, variation in neuronal soma size is often accompanied by differences in physiological activity and function [34–37]. For example, within the robust nucleus of the arcopallium (RA) in the songbird song system, neurons increase soma size during the breeding season [38] and these larger neurons have higher spontaneous firing rates than smaller neurons in the non-breeding season [39,40]. The implication then is that the granule cells of chickens probably differ in their neurophysiology from those in junglefowl. Chickens have a higher proportion of lean muscle mass compared with junglefowl [15], and if there are more (or larger) muscle fibres to innervate, this could require physiological differences in the granule cells in order to regulate locomotion [41] and other behaviours. Alternatively, there may be behavioural differences between chicken and junglefowl related to motor coordination, or even cognition, that have yet to be identified.

Overall, we conclude that cerebellar enlargement in chickens is due to a combination of intraspecific scaling rules and differential enlargement of granule cells. The extent to which the same pattern applies to other domesticated species remains to be tested. As mentioned previously, chickens are somewhat unique in that they have undergone an enlargement of a brain region, rather than the decreases observed across the majority of other species investigated [6–13]. Decreases in brain region sizes could arise from a range of different mechanisms, including changes in neuronal density, in other domesticated species, and different brain regions might vary in the extent to which neuron numbers and neuron sizes change. Determining what neuroanatomical differences occur in domesticated and wild strains at a finer level is important for determining how brain regions shrink in the domesticated brain as well as insights into some of the neural mechanisms responsible for behavioural differences between domesticated and wild strains [42].

Ethics. The study was approved by the local Ethical Committee of the Swedish National Board for Laboratory Animals (ethical permit no. Dnr 50-13) and carried out in accordance with the approved guidelines.
Data accessibility. All neuroanatomical data are available in the Dryad Digital Repository: https://doi.org/10.5061/dryad. hhmgqnkhj [43].
Authors' contributions. R.H. and A.N.I. conceived of, designed and coordinated the study; D.W. and R.H. were responsible for animal care and collecting and preparing samples; K.J.R., C.P. and F.C. processed and analysed tissue samples; K.J.R., R.H. and A.N.I. carried out statistical analyses and drafted the manuscript. All authors assisted in revising the manuscript, gave final approval for publication and agree to be held accountable for the work performed therein.
Competing interests. We declare we have no competing interests.
Funding. Funding was provided by grants to D.W. from the European Research Council (Consolidator grant FERALGEN 772874) and to A.N.I. from the Canada Research Chairs Program, Canada Foundation for Innovation, and the Natural Sciences and Engineering Research Council of Canada (NSERC), and an NSERC Undergraduate Student Research Award to K.J.R.

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
