## [Peer Review File · Royal Society Open Science]

Review History

RSOS-211002.R0 (Original submission)

Review form: Reviewer 1 (Ana Balcarcel)

Is the manuscript scientifically sound in its present form?

Yes

Are the interpretations and conclusions justified by the results?

Yes

Is the language acceptable?

Yes

Do you have any ethical concerns with this paper?

No

Have you any concerns about statistical analyses in this paper?

No

Recommendation?

Accept with minor revision (please list in comments)

Comments to the Author(s)

I enjoyed reading this work very much, and thank you for such an informative study. I had one minor adjustment to suggested, related to overall brain size comparison in the field of domestication. I am happy to provide references if you need them, but the Kruska studies have this information as well. Details in the word document attached (see Appendix A).

Review form: Reviewer 2**Is the manuscript scientifically sound in its present form?**

Yes

Are the interpretations and conclusions justified by the results?

Yes

Is the language acceptable?

Yes

Do you have any ethical concerns with this paper?

No

Have you any concerns about statistical analyses in this paper?

No

Recommendation?

Major revision is needed (please make suggestions in comments)

Comments to the Author(s)

This article reports analyses of cerebellum anatomy in domesticated white leghorn chickens and their wild progenitors, red junglefowl. This research topic is interesting and relevant because prior work had reported enlargement of the cerebellum in these chickens relative to junglefowl, in contrast to a number of different studies in other domesticated species, which generally report brain size or region size reduction compared to wild forebears. The prior study had linked this enlargement with some behavioral and genetic shifts, but had not quantified any more specific neuroanatomical measures. In the current study, the authors quantified the foliation of the cerebellar cortex, the volumes of its layers, and the numbers and sizes of Purkinje and granule cells. They found that the chicken cerebellum is essentially a scaled-up version of the junglefowl cerebellum, but with increased foliation and increased granule cell size. These results shed some light on the root causes of cerebellar enlargement in chickens relative to their wild forebears.

Overall, this study appears to be well-executed and clearly written up. I have several relatively minor comments:

- Was the histological analysis blinded (i.e., were experimenters aware whether a particular section was WLH or RJF?)
- Can you please report whether total brain:body size is significantly altered in chickens vs junglefowl? I see that the measurements are available in the table, but readers may want to know

whether there is a significant difference, in light of all the prior attention to this measure in domestication research generally.

- You find that cerebellar enlargement is due to disproportional enlargement of cells within the granule cell layer in chickens. Is it possible that increased neuropil fraction also contributes to cerebellar enlargement?

- Regarding the section of the discussion on behavioral relevance of cerebellum changes - in addition to implications for motor behavior, is it possible that there are also implications for cognition and/or affective processing?

- Only males were studied. This is clearly noted in the methods and I do not think it is a major issue, but for the sake of helping readers interpret the findings, can you please address whether there are any known sex differences in the anatomy of the avian cerebellum which might suggest that results could be different for females?

Decision letter (RSOS-211002.R0)

Dear Dr Iwaniuk

On behalf of the Editors, we are pleased to inform you that your Manuscript RSOS-211002 "The cerebellar anatomy of Red Junglefowl and White Leghorn Chickens: insights into the effects of domestication on the cerebellum" has been accepted for publication in Royal Society Open Science subject to minor revision in accordance with the referees' reports. Please find the referees' comments along with any feedback from the Editors below my signature.

Please submit your revised manuscript and required files (see below) no later than 7 days from today's (ie 16-Jul-2021) date. Note: the ScholarOne system will 'lock' if submission of the revision is attempted 7 or more days after the deadline. If you do not think you will be able to meet this deadline please contact the editorial office immediately.

on behalf of Professor Marcelo Sanchez (Associate Editor) and Kevin Padian (Subject Editor)
 openscience@royalsociety.org

Reviewer comments to Author:

Reviewer: 1

Comments to the Author(s)

I enjoyed reading this work very much, and thank you for such an informative study.

I had one minor adjustment to suggested, related to overall brain size comparison in the field of domestication. I am happy to provide references if you need them, but theKruska studies have this information as well. Details in the word document attached.

Reviewer: 2

Comments to the Author(s)

This article reports analyses of cerebellum anatomy in domesticated white leghorn chickens and their wild progenitors, red junglefowl. This research topic is interesting and relevant because prior work had reported enlargement of the cerebellum in these chickens relative to junglefowl, in contrast to a number of different studies in other domesticated species, which generally report brain size or region size reduction compared to wild forebears. The prior study had linked this enlargement with some behavioral and genetic shifts, but had not quantified any more specific neuroanatomical measures. In the current study, the authors quantified the foliation of the cerebellar cortex, the volumes of its layers, and the numbers and sizes of Purkinje and granule cells. They found that the chicken cerebellum is essentially a scaled-up version of the junglefowl cerebellum, but with increased foliation and increased granule cell size. These results shed some light on the root causes of cerebellar enlargement in chickens relative to their wild forebears.

Overall, this study appears to be well-executed and clearly written up. I have several relatively minor comments:

- Was the histological analysis blinded (i.e., were experimenters aware whether a particular section was WLH or RJF?)
- Can you please report whether total brain:body size is significantly altered in chickens vs junglefowl? I see that the measurements are available in the table, but readers may want to know whether there is a significant difference, in light of all the prior attention to this measure in domestication research generally.
- You find that cerebellar enlargement is due to disproportional enlargement of cells within the granule cell layer in chickens. Is it possible that increased neuropil fraction also contributes to cerebellar enlargement?
- Regarding the section of the discussion on behavioral relevance of cerebellum changes - in addition to implications for motor behavior, is it possible that there are also implications for cognition and/or affective processing?
- Only males were studied. This is clearly noted in the methods and I do not think it is a major issue, but for the sake of helping readers interpret the findings, can you please address whether there are any known sex differences in the anatomy of the avian cerebellum which might suggest that results could be different for females?

===PREPARING YOUR MANUSCRIPT===

one version identifying all the changes that have been made (for instance, in coloured highlight, in bold text, or tracked changes);
 a 'clean' version of the new manuscript that incorporates the changes made, but does not highlight them. This version will be used for typesetting.

===PREPARING YOUR REVISION IN SCHOLARONE===

- Any electronic supplementary material (ESM).
- If you are requesting a discretionary waiver for the article processing charge, the waiver form must be included at this step.
- If you are providing image files for potential cover images, please upload these at this step, and inform the editorial office you have done so. You must hold the copyright to any image provided.
- A copy of your point-by-point response to referees and Editors. This will expedite the preparation of your proof.

- Ensure that your data access statement meets the requirements at <https://royalsociety.org/journals/authors/author-guidelines/#data>. You should ensure that you cite the dataset in your reference list. If you have deposited data etc in the Dryad repository, please only include the 'For publication' link at this stage. You should remove the 'For review' link.
- If you are requesting an article processing charge waiver, you must select the relevant waiver option (if requesting a discretionary waiver, the form should have been uploaded at Step 3 'File upload' above).
- If you have uploaded ESM files, please ensure you follow the guidance at <https://royalsociety.org/journals/authors/author-guidelines/#supplementary-material> to include a suitable title and informative caption. An example of appropriate titling and captioning may be found at https://figshare.com/articles/Table_S2_from_Is_there_a_trade-off_between_peak_performance_and_performance_breadth_across_temperatures_for_aerobic_scope_in_teleost_fishes_/3843624.

Author's Response to Decision Letter for (RSOS-211002.R0)

See Appendix B.

Decision letter (RSOS-211002.R1)

Dear Dr Iwaniuk,

I am pleased to inform you that your manuscript entitled "The cerebellar anatomy of Red Junglefowl and White Leghorn Chickens: insights into the effects of domestication on the cerebellum" is now accepted for publication in Royal Society Open Science.

You can expect to receive a proof of your article in the near future. Please contact the editorial office (openscience@royalsociety.org) and the production office

(openscience_proofs@royalsociety.org) to let us know if you are likely to be away from e-mail contact -- if you are going to be away, please nominate a co-author (if available) to manage the proofing process, and ensure they are copied into your email to the journal. Due to rapid publication and an extremely tight schedule, if comments are not received, your paper may experience a delay in publication.

on behalf of Professor Marcelo Sanchez (Associate Editor) and Kevin Padian (Subject Editor)
openscience@royalsociety.org

Appendix A

This is a valuable study that adds much to what is known about brain anatomy in domestic versus wild organisms, beyond simply overall size. The methods are relatively novel to the study of brains under domestication and that is a great contribution. The narrative is clear and well-written.

My only concern is that the comparison of overall brain size is absolute. Almost exclusively, the comparisons of brain size between wild and domestic animals are relative, i.e., corrected for body size (allometric comparisons). This is highly relevant to consider. In Table 1, the absolute body size increase is much larger than the absolute brain size increase in chicken versus RJF. It would be important to make this allometric adjustment, in order to compare with results in other wild/domestic pairs. Otherwise, I do not think this is actually contradicting the trend seen under the “domestication syndrome”. However, if this is adjusted in the Introduction and throughout, this aspect of the study would be more reliable and comparable to other works.

Minor edit:

Small typo line 53/54 in “stain”, should be “strain”.

Appendix B

University of
Lethbridge

Department of Neuroscience
4401 University Drive
Lethbridge, Alberta, Canada T1K 3M4

Canadian Centre for Behavioural Neuroscience
Phone 403.394.3900
Fax 403.329.2775
<http://ccbn.ulethbridge.ca>
<http://www.ulethbridge.ca/fas/neur>
ccbn@ulethbridge.ca

13 September 2021

To Whom it May Concern,

I wish to resubmit the attached manuscript entitled: “Quantitative differences in the cerebellar anatomy of Red Junglefowl and White Leghorn Chickens: insights into the effects of domestication on the cerebellum.” for publication in Royal Society Open Science.

Due to a series of miscommunications, I did not provide a revised version of the manuscript by the stated deadline, so I am asking this to be treated as a resubmission.

All of the research reported herein was approved by Linköping University and the University of Lethbridge and adheres to all relevant ethical regulations. A statement to this effect is included in the manuscript. In addition, all of the research reported in this paper is original and is not being considered for publication elsewhere.

Recently, there has been a renewed interest in studying how the process of domestication and selection for different breed traits results in changes in the anatomy and neurochemistry of the brain. Chickens are unusual among domesticated species in that they are one of a handful of examples where a brain region has become enlarged relative to the wild type. More specifically, chickens have larger cerebella than the ancestral junglefowl. In our manuscript, we test whether the difference in cerebellum size between junglefowl and chickens is due to more neurons, larger neurons or a combination of the two. Our data show that, in general, neuron numbers are proportionally similar between junglefowl and chickens, but chickens have relatively larger granule cells. This represents the first study to quantify neuroanatomical differences between a domesticated strain and wild-type of any animal and demonstrates that the effects of domestication/line breeding do not necessarily involve changes in neuron numbers.

Because the study of domestication crosses disciplines, our results will be of broad interest to neuroscientists, organismal biologists and evolutionary biologists and is well suited to Royal Society Open Science. We hope that the editor and reviewers agree with this assessment and look forward to hearing from you soon.

Sincerely,

Andrew Iwaniuk